# Trends in Workplace Injuries in Slovak Forest Enterprises

**DOI:** 10.3390/ijerph16010141

**Published:** 2019-01-07

**Authors:** Miloš Gejdoš, Mária Vlčková, Zuzana Allmanová, Žaneta Balážová

**Affiliations:** 1Department of Forest Harvesting, Logistics and Ameliorations, Faculty of Forestry, Technical University in Zvolen, 960 01 Zvolen, Slovakia; gejdos@tuzvo.sk (M.G.); zuzana.allmanova@tuzvo.sk (Z.A.); 2Institute of Foreign Languages, Technical University in Zvolen, 960 01 Zvolen, Slovakia; zaneta.balazova@tuzvo.sk

**Keywords:** workplace injuries, work safety, health and safety, work errors forestry, injury rate

## Abstract

The aim of the paper is to analyse the effect of key factors affecting the risk of workplace injuries and to identify the most common workplace accidents regarding injured body parts with respect to anthropometric data measurements of the population. Data associated with workplace accidents over the years 2000–2016 were drawn from the records of the state enterprise Forests of the Slovak Republic, situated in Banská Bystrica. Gathered data were processed and entered into the database complemented by the data on accidents of the self-employed working in the forestry industry. A total of 1874 workplace accidents in the state enterprise were recorded and statistically evaluated during the analysis period. A method for contingency table was used to analyse correlation between qualitative (categorical) variables in the dataset. A Poisson regression model was used to determine the injury rate. Forest harvesting is considered the most risky phase of the process of harvesting, processing, and transport. The highest number of workplace accidents (31.8% of all recorded workplace accidents) occurred during the forest harvesting phase during the analysis period. Timber skidding, with 16% of recorded accidents, was the second highest-risk phase. The workplace injury rate in the forest industry in Slovakia decreased over the course of the years 2000–2016. Head and facial injuries were those with the highest rate (67.1% injuries of these body parts) during the phase of harvesting and skidding.

## 1. Introduction

The scope of human physical and mental abilities is limited. Long-term uncomfortable or painful feelings can cause damage to organisms due to a significant weakening in performance, with a higher injury risk [1]. The forestry sector is one of the industries of the national economy with the highest injury risk. The specific working conditions in forests, involving the type of terrain, weather conditions, and the use of power tools and heavy machinery, make work hazardous, with a certain risk of serious injuries [1,2,3,4]. The risk of work-related injuries is very high in forestry jobs in general [5,6]. In order to create healthy and safe working environments, processes, and conditions, accident investigations can reveal information about causation that can be used to develop injury prevention and protection programmes.

In order to create health and safe working environment, real accidents have to be analysed and subsequently, a system of preventive measures has to be developed. Methods to protect health are constantly evolving. Forestry generates many jobs, which are offered mostly by private companies in the case of Slovakia. Several activities carried out on forest estates are managed by private enterprises that underestimate the risks to health and safety that can arise. When health and safety legislation and requirements are not met, the health and lives of workers are endangered. Work-related health issues in small enterprises are recorded in many countries, both developing and developed. Some examples include South Africa, Sweden, Slovakia, and Croatia [7,8,9,10].

Many previous studies have focused on injury incidence and prevalence to underscore the need for great safety in the workplace. The authors of [11,12] state that an analysis of injury rate can provide complete information about changes in injury rate depending on a specific prognostic variable. Further results can be gathered using Poisson regression models.

Employees in forestry, especially those exposed to harmful factors for at least six years, appear to suffer from work-related injuries. Noise, vibration, extreme temperatures, dust, exhaust fumes (especially carbon monoxide), and fixed or constrained body positions are considered to be harmful to health. [13]. Working with chainsaws, the most commonly used tool by forest workers in Slovakia, can be considered one of the most dangerous activities in forestry.

The aim of the paper is to analyse work phases in forestry that may be associated with risk of injury. Moreover, the research is focused on determining the significant factors affecting the risk of damage to the health in the forestry sector.

## 2. Materials and Methods

### 2.1. Data Collection and Processing

Data associated with workplace injuries were drawn from the records of the state enterprise Forests of the Slovak Republic situated in Banská Bystrica. Gathered data were processed and entered into the database, complemented by data on injuries of the self-employed working in forestry. Questionnaires about the accidents were delivered to various forest enterprises in order to gather relevant information about circumstances and causes of workplace accidents. Workplace accidents were identified with accordance with “European Statistics on Accidents at Work” (ESAW) [14] in order to create a dataset. Circumstances and causes of workplace accidents in Slovakia were determined using the authors’ classification based on the Decree of Ministry of Labour, Social Affairs and Family of the Slovak Republic No. 500/2006 Coll. on establishment of a template for reported occupational accident in accordance with the unified statistical reporting system in the European Union [15]. A dataset of workplace accidents was created over the years 2000–2016. During the analysis period, 1874 workplace accidents in forest enterprises in Slovakia were recorded.

### 2.2. Contingency Tables

A method for contingency tables was used to analyse the correlation between qualitative (categorical) variables in a dataset consisting of workplace injuries.

Investigation of the correlation between two variables is based, similarly to a one-dimensional statistical dataset, on finding frequencies mentioned in contingency tables. The research methodology follows that of several works [16,17,18].

For the case of the two categorical variables A and B, where the classes of variable A are A_1_, A_2_, A_3_, … A*_k_* and the classes of variable B are B_1_, B_2_, B_3_ … B*_m_*, a *k* × *m* contingency table as illustrated in Table 1 is formed.

The degree of correlation between two categorical variables A and B is measured following the comparison of the observed frequencies in individual classes of the contingency table *n_ij_* to the expected frequencies *nʹ_ij_*, when the variables are independent. The expected variables are calculated using Formula (1):(1)$${n^{\prime }}_{ij}=\frac{n_{i}\cdot n_{j}}{n}$$

They are calculated as the product of the corresponding frequencies (*n_i_* for the variable A and *n_j_* for the variable B) divided by the size of dataset *n*.

The chi-squared test of independence was used to determine whether the occurrence of serious workplace accidents during the phase of harvesting and skidding is statistically significant.

Testing the hypothesis on the significance of expected dependence is based on the calculation of *χ*^2^ (chi-squared) defined by Formula (2):(2)$$\chi^{2}=\overset{k}{\underset{i=1}{\sum }}\overset{m}{\underset{j=1}{\sum }}\frac{\left(n_{ij}-{n^{\prime }}_{ij}\right){}^{2}}{{n^{\prime }}_{ij}}$$

*χ*^2^ is calculated in the contingency table, where besides observed frequencies *n_ij_* (cell of the table), the calculated expected frequencies *nʹ_ij_*, and even the differences (*n_ij_* − *nʹ_ij_*) are displayed.

Expected frequencies (*nʹ_ij_*) must be calculated for cells of the table without observed frequencies. The value of these frequencies is *χ*^2^ (Equation (3)):(3)$${n^{\prime }}_{ij}=\frac{{\left(n_{ij}-{n^{\prime }}_{ij}\right)}^{2}}{{n^{\prime }}_{ij}}$$

The results gathered using the formula are relevant when the size of dataset is *n* > 40. In case that 20 < *n* < 40 and some of expected frequencies *nʹ_ij_* are less than 5, the class with the mentioned frequency must be combined with two or more adjacent classes of variable A or B. The mentioned method cannot be used in the case of the size of dataset is *n* < 20.

Testing the null hypothesis (*H*_0_) on the independence of variables A and B follows *χ^2^*:(4)$$H_{0}\;:\;n_{ij}-{n^{\prime }}_{ij}=0,\;or\;\overset{k}{\underset{i=1}{\sum }}\overset{m}{\underset{j=1}{\sum }}\left(n_{ij}-{n^{\prime }}_{ij}\right)=0$$

Small values of *χ*^2^ confirm *H*_0_; on the other hand, large values of *χ*^2^ reject it.

The test statistics have an approximately *χ*^2^ distribution only if less than 20% of *nʹ_ij_* < 5 ∀ *i*, *j*. If *χ*^2^ > *χ*^2^
_(*k*−1) (*m*−1)_ (α), the hypothesis on the independence of variables A, B is rejected.

Critical values (*k* − 1) (*m* − 1) (α) are percentage points, and (*k* − 1) (*m* − 1) is the number of degrees of freedom.

Sometimes *χ*^2^ is called the likelihood ratio. The value of *χ*^2^ provides information whether correlation between variables A and B can be considered to be statistically significant or not. However, it does not provide information on the degree of dependence of the variables. The degree of dependence can be measured using the coefficient of association of two categorical variables A and B. It is calculated using the Formula (5) [18]:(5)$$r_{\mathrm{AB}}=\sqrt{\frac{\chi^{2}}{n\sqrt{\left(k-1\right)\cdot \left(m-1\right)}}}$$

### 2.3. Injury Rate Analyses

The aim of the injury rate analysis is to make decision about the effect of the commonly used prognostic variables on the injury rate. Moreover, the association between the prognostic variables and injury rate is studied as well [19]. The analysis is carried out using the Poisson regression model. The model is often used to analyse the data related to age group in epidemiology [20,21,22,23] and it is mentioned in several works [19,24,25].

Statistical models provide a possible solution to the problem. Poisson regression models provide a standard framework for more complex statistical analyses of injury rate. However, a detailed insight into the impacts of many variables is possible as well. The injury rate is estimated with regard to the number of accidents (*d*) divided by the amount of produced wood (1000 m^3^) (*N*),
injury rate = *d*/*N*, (number of accidents/1000 m^3^)(6)

Data gathered in the year 2017 could not be included in the analysis because data associated with the wood produced by the Slovak forest industry in that year were not available. Data associated with wood production over the previous years were drawn from the Green report issued by the Ministry of Agriculture and Rural Development of the Slovak Republic [26].

The number of injuries is predicted using the Poisson distribution with the mean *µ_d_* = *N* × *λ*, where *λ* is the incident injury rate. The average number of injuries, injury rate is expressed by parametric function of prognostic variables. As the Poisson distribution assumes that the meaningful value in distribution is greater than zero, this function of prognostic variables is often limited by values of a function greater than zero. The following logarithmic-linear model is usually used to model the injury rate *λ* as a function of a set of prognostic variables *X*_1_, … *X_p_*:
log (*λ*) = log (*µ_d_*/*N*) = *β*_0_ + *β*_1_*X*_1_ + … + *β_p_ X_p_*(7a)
or
log (*µ_d_*) = *β*_0_ + *β*_1_*X*_1_ + … + *β_p_ X_p_* + log (*N*)(7b)
where *β* is a regression coefficient of average number of injuries of *i*-*th* prognostic variable *X* (*i* = 1 … *n*). Thus, the log (of rate) is a linear function of prognostic variables. In the language of generalised linear models (GLMs), the usual specification of the linear component is due to the inclusion of the term log (*N*). This term is called the “offset” [27,28]. In the log function (Formula 7b) the average number of injuries is expressed using the linear combination of prognostic variables, while log (*N*) is expressed by fixed-value coefficient. The analyses relate to the amount of produced wood. The number of employees (full-time as well as part-time workers) can be used as an alternative to the produced wood (1000 m^3^).

If the variance and the mean are equal, this is the condition to select the Poisson distribution for the number of injuries (*d*). Multiple linear regression with homogeneity of variances is based on the Poisson distribution.

If the variance and the mean are not equal, standard errors relating to parameter proposal are incorrect. The condition is usually broken when the variance is larger than the mean, the so-called over-variance. Methods dealing with over-variance are mentioned in the work [28]. The relevancy test for Poisson regression models of both the variance and the mean is determined in the *χ^2^* distribution used to compare the observed and expected frequencies [27]. A comparison of statistical models based on measuring the model discrepancy and variance is also used for qualitative evaluation of model adequacy. If the variance and the degrees of freedom are approximately equal, the model is considered to be adequate.

Many important aspects have to be taken into consideration when the models are used, for example, whether models are specified correctly, the determination of prognostic variables, and appropriate methods for distribution. Additional analyses have to be carried out in order to determine the suitability of the regression model for making predictions. Moreover, the data quality must be evaluated (reliability, accuracy, etc …) as well as random factors related to the analyses of injury rate resulting from the various source of data.

## 3. Results

### 3.1. Injuries in Forestry Operations

Timber harvesting is the most risky phase of the process of harvesting, processing, and transport (Figure 1). The highest number of workplace accidents (31.8% of all recorded workplace accidents) occurred during the mentioned phase in the analysis period, due to motor-assisted technology with chainsaws in the timber harvesting phase. Timber skidding, with 16% of recorded accidents, was the second highest-risk phase. The timber harvesting phase is the riskiest phase. Forty-eight percent of fatal injuries and 26% of serious injuries leading to permanent damage to health were recorded in the forest industry during the analysis period. Seventy-five percent of all serious or fatal injuries and 48% of all workplace accidents mentioned in the database occurred during the phase of harvesting and skidding.

Observed and expected frequencies related to fatal injuries and serious workplace injuries leading to permanent damage to health during the phase of harvesting and skidding and also during other phases and activities with recorded workplace injuries are shown in a contingency table (Table 2). The null hypothesis was tested:

**Hypothesis** **1.**
*The difference in the occurrence of fatal injuries and serious workplace injuries leading to permanent damage to health during the phase of harvesting and skidding and also during other phases and activities is not statistically significant.*


*χ*^2^ = 47.4400Degree of freedom (DF): 1

At the level of significance *α* = 0.05 with the degree of freedom 1, the value of chi-squared is 3.8. The calculated *χ^2^* is bigger than *χ*^2^_1(0.05)_ and the null hypothesis can be rejected with 95% confidence. The difference in the occurrence of fatal injuries and serious workplace injuries leading to permanent damage to health during the phase of harvesting and skidding and also during other phases and activities is statistically significant. More fatal injuries than the “expected dependence” occurred during the phase of harvesting and skidding. The risk of fatal injuries recorded in the mentioned phases is higher than in all other phases and activities in the forest industry.

Due to the coefficient of association of *r*_AB_ = 0.32, there is a low positive relationship.

### 3.2. Injury Rate

The injury rate in the forest industry over the years 2000–2016 was decreasing. The trend in the injury rate for the forest industry on 1000 m^3^ of produced wood in the years 2000–2016 is illustrated in the graph (Figure 2).

A simple one-dimensional Poisson regression model can be used to analyse whether the trend in the injury rate is a function of one hypothetical variable—the calendar year. The number of workplace injuries and the amount of produced wood in the Slovak forest industry in 1000 m^3^ were primary data in the first model. The effect of other prognostic variables was not taken into consideration. It is expressed using the formula:(8)$$\ln\left(\lambda_{year}\right)=b_{0}+b_{1}\left(year\right)$$where *λ_year_* is an injury rate or each year.

A regression model (Figure 3) described using Formula (8) (hereinafter Model (8)), means that ln (injury rate) should be dependent on the year (*b*_1_). Results of the regression analysis are mentioned in Table 3.

Coefficients used in Model (8) were *b*_0_ = 327.0145 and *b*_1_ = −0.1617, with the standard error of s.e._(*b*0)_ = 35.7358 and s.e._(*b*1)_ = 0.0178. As is shown in the model, the injury rate decreases by 0.16 units per year after logarithmic transformation. Expected change in the number of injuries was exp (−0.1617) = 0.8510 from year to year. Therefore, the injury rate decreases approximately by 14.9% (= 100 × (1 − 0.8510)%) annually. An approximate 95% confidence interval for the parameter which is changed yearly is −0.1617 ± 1.96 (0.0178) = (−0.1264; −0.1969). Seeing that *b*_1_ = 0 is not in this interval, a significant change in the primary observed injury rate per year is expected.

### 3.3. Injured Body Parts

Forestry jobs have a risk of permanent damage to health or ability to perform manual tasks. Therefore, an analysis of injured body parts has to be carried out. According to many authors, the anthropometric parameters of the Slovak adult population have been increasing to date [29,30,31]. That is why the risk of workplace accidents in several forest operations has been affected as well. The overview of the number of injuries related to individual body parts occurring in forest industry in the years 2000–2016 is shown in Figure 4.

The most common parts of the body affected by workplace accidents are legs—all parts of legs (42.5% of all injuries). Leg injuries are not usually life-threatening. Head and facial injuries are those with the highest rate (67.1% injuries of these body parts) during the phase of harvesting and skidding. Moreover, these injuries can result in life-threatening damage. Injuries of the whole body or several body parts at the same time occur most often (54.2%) in the phase of harvesting and skidding. The mentioned results correspond to the high rate of fatal injuries in the phase of forest harvesting and timber skidding (3.1), as injuries to these body parts are most often life-threatening.

## 4. Discussion

A non-significant decrease in the injury rate over the years 2000–2016 was determined in the analysis of the workplace injury rate in the Slovak forest sector. Grzywinski et al. [32] analysed workplace accidents in Szczecinek (NW Poland) legally administered by the Regional Directorate of State Forests in the years 1990–2009. A total of 10,420 workplace accidents and an injury rate of almost 15 injuries per 1000 employees were recorded. Following the analysis, it seems the privatisation in the forestry industry did not affect the number of workplace accidents significantly.

Benavides et al. [33] carried out an analysis in Spain, finding that the injury rate of full-time workers (44.9 per 1000 employees in the year 2002) was lower than the injury rate of seasonal workers (120.6 per 1000 employees in the year 2002). This is influenced by the quality of professional training as well as the work experience of full-time workers. In [34], a decrease in fatal injury rate in agriculture, game keeping, and forestry in Arkhangelsk region from 43.9 per 1000 employees in the year 1996 to 20.8 per 1000 employees in the year 2007 is mentioned. The authors of [35] compared the fatal injury rates in Australia and the United States over the years 1989–1992 and in New Zealand over the years 1985–1994. The average fatal injury rate in New Zealand was 4.9/100,000, while in Australia it was 3.8/100,000, and in the United States it was 3.2/100,000. In the course of the years 1975–1984, the death rate of forest workers in New Zealand was 8.0 per 100,000 employees per year.

Suchomel [36] carried out the analysis of workplace accidents in the process of forest harvesting over the years 1984–1993. The position of individual phases in the process of harvesting, processing and transport was as follows: forest harvesting—32%, timber skidding—29%, work on the skidway—9%, timber hauling—8%, work at the conversion depot—6%, repairs and maintenance—15%, and internal control—1%. Following the overview, almost the same order of the phases as during the analysis period 2000–2016 can be seen. Small differences result from subjective as well as objective factors. In the work [37] the analysis of Sweden is carried out. The results show that serious workplace accidents are very common in Sweden and the death rate of self-employed in forestry is 7% of the total number of workers killed in work-related activities recorded by the employment office. Falling trees are the main cause of fatal or life-threatening injuries. Dangerous practises and procedures at work result in injuries when using work tools (e.g., portable chainsaws). Therefore, quality training as well as skill improvement must be offered to line workers to equip them to deliver health and safety to the workplace.

When evaluating specific injured body parts in the analysis of workplace accidents, the authors [37] found out that a knee is the most commonly injured part of the body in the process of forest harvesting. The results of our analysis show that leg injuries occurred most often during harvesting as well (258). This represents approximately 14% of all recorded injuries. An injury rate of 28% is mentioned in [38], a rate of 14% of all injuries in [36], and a rate of 19% (excluding foot and ankle injuries) in [1].

## 5. Conclusions

Forest harvesting and timber skidding represent operations with a high risk of injury. Working with chainsaws, the most commonly used tool by forest workers in Slovakia, can be considered one of the most dangerous activities in forestry. In order to create a healthy and safe working environment, real accidents have to be analysed and subsequently, a system of preventive measures has to be developed [39].

The self-employed must be the focus of attention. Making money is more important for them than following safety instructions and advice. Their equipment is not in accordance with regulations, and they do not wear protective clothing made for forestry workers seeking the best combination of safety and optimal ergonomics in daily work. They underestimate and break the general norms for ensuring the safety, health, and welfare of workers and do so intentionally. Minor injuries not threatening their lives or not leading to permanent damage to health are not recorded and the labour inspectorate is not informed about them. Forest jobs are also commonly offered to enterprises not putting equal emphasis on qualification, training, or skills. Workers have only poor information about doing enough to protect their health and keep them safe at work. The job of forest operators is a physically demanding job associated with health risk. The abovementioned facts are often ignored by the self-employed, who often work alone—for themselves. In case of workplace accidents there is nobody to arrange first aid or any other medical treatment they need on time. It is necessary to act and think quickly to avoid delayed treatment during an emergency, although this can be seen as ineffective or even harmful. Moreover, they often do not have health insurance coverage from commercial health insurance due to lack of finances and consequently, serious injury can result in severe financial hardship.

In practise, measures for prevention and control can be inspired by the results of the analysis. Following the analysis of the records related to workplace injuries and occupational diseases, the risk to health during the process of harvesting, processing, and transport can be evaluated. Changes in forest operations, especially in the phase of forest harvesting and timber skidding, aimed at technology innovation, environmentally-friendly procedures, and protection of the employees from workplace dangers need to be made. The education and training of forest operators should be focused on the safe use of a chainsaw, the proper use of personal protective equipment, and also on maintaining the good posture in order to prevent workplace injuries caused by fatigue. Mechanical and physical risk prevention must be supported in small and medium-sized enterprises (there are currently very weak prevention measures). In the case of a decrease in workplace injuries, the social insurance contribution paid by an enterprise should be lower. This is a method to motivate employers to follow requirements in order to provide a safer environment in the workplace. In order to cope better with health and safety issues for the self-employed, further safety legislation and laws must be implemented.

Even though the results of the analysis show a decrease in injury rate considering an increase in forest production, misinterpretation is a common problem when using statistical information, especially data related to the self-employed. Due to a high number of fatal and serious injuries in the process of forest harvesting and timber skidding, there is widespread recognition of the importance of good education and training of forest operators. Great emphasis must be placed on the use of proper equipment and technology as well as safety processes and procedures to minimise the risk of injury in the workplace.

## Figures and Tables

**Figure 1 ijerph-16-00141-f001:**
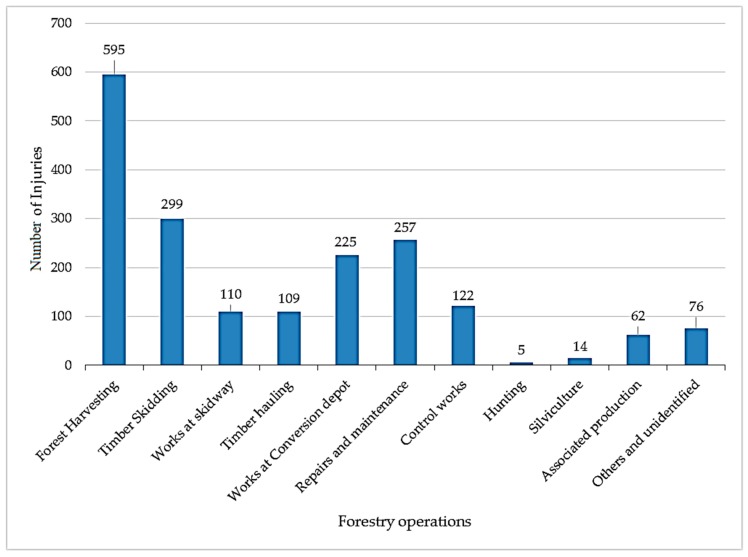
Workplace injuries in Slovak forestry over the years 2000–2016.

**Figure 2 ijerph-16-00141-f002:**
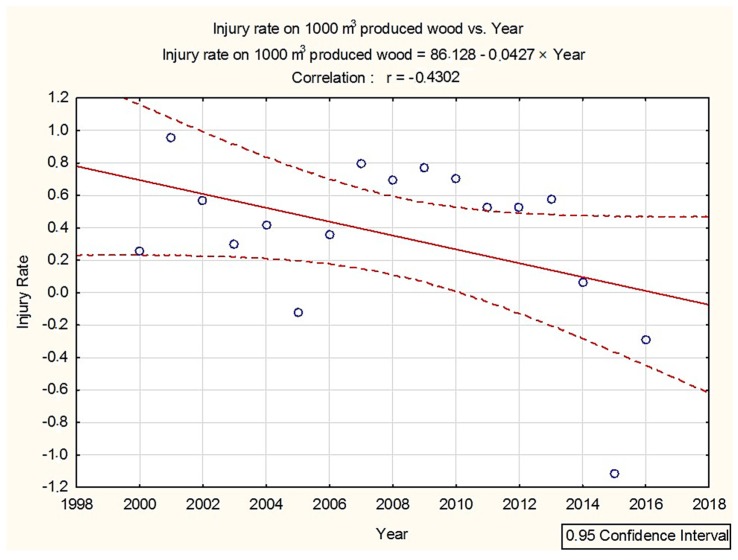
Trend in the injury rate on 1000 m^3^ in the Slovak forestry sector over the years 2000–2016.

**Figure 3 ijerph-16-00141-f003:**
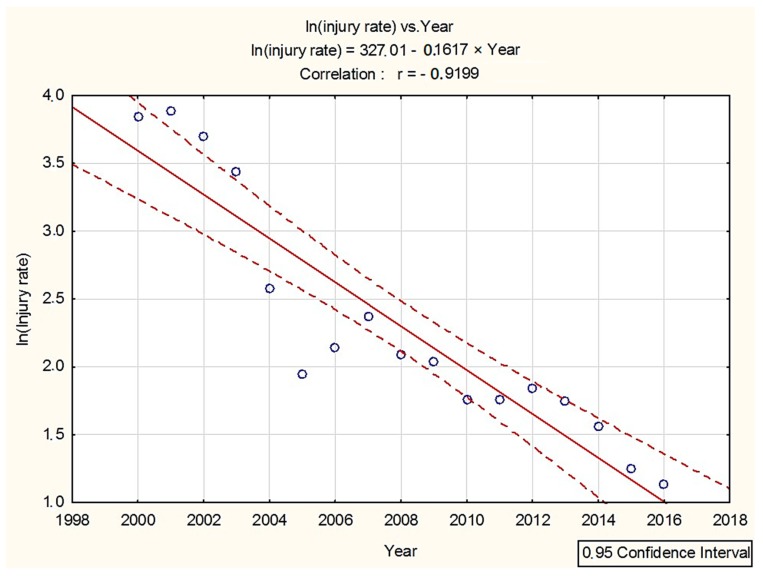
Trend in the injury rate in the Slovak forestry sector over the years 2000–2016 after calculation with Model (8).

**Figure 4 ijerph-16-00141-f004:**
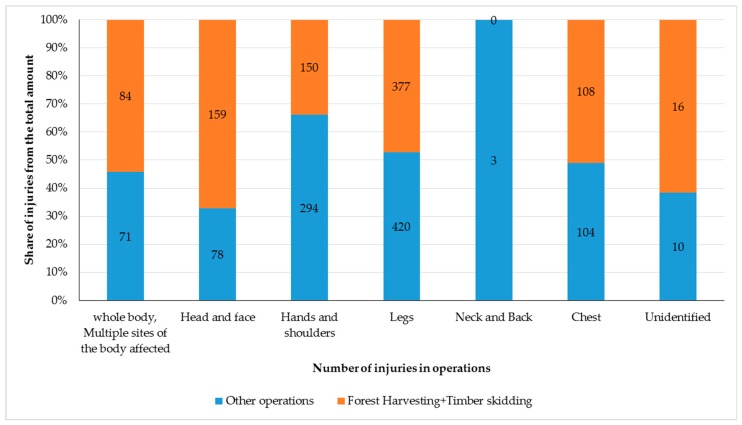
Injured body parts due to forestry operations in Slovak forestry over the years 2000–2016.

**Table 1 ijerph-16-00141-t001:** Template of contingency table.

Variable B		Classes of Variable B						Total
Variable A		B_1_	B_2_	…	B*_J_*	…	B*_m_*	
Classes of Variable A	A_1_	*n_11_*	*n_12_*	…	…	…	*n_1m_*	*n_1A_*
	A_2_	*n_21_*	*n_22_*	…	…	…	*n_2m_*	*n_2A_*
	.	.	.		.		.	.
	A*_i_*	*n_i1_*	*n_i2_*	…	*n_iJ_*	…	*n_im_*	*n_JA_*
	.	.	.		.		.	.
	A*_k_*	*n_K1_*	*n_K2_*		*n_KJ_*		*n_Km_*	*n_kA_*
Total		*n_B1_*	*n_B2_*	…	*n_BJ_*	…	*n_Bm_*	*N*

Source: Authors’ compilation.

**Table 2 ijerph-16-00141-t002:** Contingency table for the occurrence of fatal injuries (A-injuries) and serious injuries (B-injuries) in forest harvesting and timber skidding operations.

Phase	Injuries/Abundance	A-Injuries	B-Injuries	Σ
Forest harvesting + timber skidding	Real injuries	96	126	222
	Expected abundance	63	159	
Other	Real injuries	32	199	231
	Expected abundance	65	166	
Σ		128	325	453

Source: Authors’ compilation.

**Table 3 ijerph-16-00141-t003:** Simple regression results with dependence variable (injury rate in forestry).

Simple Regression Results with Dependence Variable: C (Injury Rate) R = 0.9199, R^2^ = 0.8463, Edited R^2^ = 0.8360, F (1,15) = 82.5660, *p*						
***N* = 17**	**B**	**Standard Error from *b***	***b***	**Standard Error from *b***	***T* (15)**	***p*-value**
Absolute term			327.0145	35.7358	9.1509	0.0000
Year	−0.9199	0.1012	−0.1617	0.0178	−9.0865	0.0000

Source: Authors’ compilation.

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
