# Peer review of "Trends in Workplace Injuries in Slovak Forest Enterprises"

_ijerph, 2019, doi:10.3390/ijerph16010141_

Round 1
Reviewer 1 Report
Overall
The paper is not well structured and a deep revision is necessary. Some parts must be completely rewritten and some words are not appropriate. Also the structure of the sentences must be revised.
The number of the decimals must be uniform along all the text.
Table and figure inside titles must be avoided.
The references must be shortened (too much repetitions and/or not pertinent papers)
Introduction
Incorrect words are present at lines 32, 34, 37
You start referring to the risks in forestry sector (lines 33-42), then you cite not pertinent works ([7-10],line 45).
Lines 46-49: not clear, please explain better
Line 52: local vibration. What is this?
In the introduction you mix the forest sector with other human activities, increasing the confusion in the mind of the reader. Beside this, the goal statement of the paper is not clear.
Material and method
Lines 116-117: reduce the 7 citations to one, max 2: this is not a work on epidemiology.
Line 121: write better. What does it mean m3 (N)?
If data gathered in the year 2017 cannot be included in the analysis, this year must be avoided since the beginning (line 71).
Line 128: As the Poisson distribution assumes that, the mean is greater than zero, the function of prognostic variables is often limited and the values of a function are greater than zero. Not clear.
Line 130: Following logarithmic-linear model is usually used to express the rate λ as a function of a set of prognostic variables X1,. . . . . Xp. Please, explain better.
Line 134: The acronym of the generalized linear models is GLMs, not GLIMs.
Line 136: This term is called the “offset”. [27-28]. One citation is sufficient.
Line 138: invariable coefficient … What does it mean?
Line 140: wood in thousand.m3. ????? Not scientific
Line 140: However, the risk of part-time workers cannot be compared to the risk of full-time workers. And then?
Line 142: The variance and the mean are equal is a condition to select Poisson distribution for the number of injuries (d). ??? Not clear.
Line 143. Multiple linear regression with homogeneity of variances is based on the distribution. Not clear
Line 144-149: Absolutely not clear. Explain better
Results
Line 160: Timber harvesting is considered the most risky phase of the process of harvesting, processing and transport (Fig. 1). If this is a result of a data collection and not of a personal perception, the term ‘considered’ is not correct. Moreover, if I look the figure 1, I see the forest harvesting as the highest risky operation.
Line 163: Timber harvesting phase is one of the riskiest phases. ??? Not clear. Moreover, it not correct.
Line 164: 48 % of fatal injuries and 26% of serious injuries leading to permanent damage to health were recorded in the forest industry during the analysis period. ??? Again: the forest sector is or is not the objective of the investigation?
Lines 170-178: to be moved in material and method.
Line 189: More fatal injuries than was the expected value occurred during the phase of harvesting and skidding. Explain better
Line 192: Due to the coefficient of association of rAB = 0.32 there is a moderate positive relationship. Very, very questionable…
Figure 2: Please move out the text of the title of the figure and explain it in the paper.
Line 199: thousand.m3. Not scientifically correct.
Line 203: In this model, it is supposed that the injury rate is the same for all forest workers regardless of the gender and age. Questionable sentence. It has been studied that in the forestry activities the injury rate is related to the two mentioned variables.
Figure 3: as written for figure 2.
Line 209 (Figure 3 caption): logarithmic transmission????
Line 211: dependence variable: not used
Table 3 is not well structured. Please avoid titles inside the tables and the figures, but move out in the text.
Line 213: Coefficients of creating the model????
Line 213-219: rewrite and explain better
Line 223: anthropometric data of the Slovak adult population has been increasing. What does it mean?
Line 228: Injured body part in terms of operations?????
Line 229: (48 % of all recorded injuries in forestry during the analysis period). Already written. Avoid repetitions.
Line 230: Following the results ????
Line 232: Leg injuries are not usually life-threatening. Don’t true. It depends.
Line 229-238: Twisted sentences… difficult to understand…
Discussion
The discussion chapter is a comparison between the current work and others concerning the same topics. Here we find comments on other works (especially from line 246 to line 255), not linked to this paper. Moreover, some of the cited papers do not relate to the forest operations. There are two options: remove these citations or move them to the introduction chapter.
All the chapter is confused and must be completely rewritten.
Conclusion
Conclusions are too long and must be shortened, also for avoiding confusion.
Line 277: Fatal injuries or serious injuries leading to permanent damage to health can occur, whereby, injuries are difficult to predict or avoid. Not understandable
In the conclusion chapter you mention the danger when using chainsaws, but in the results you never cited the machines. Conclusion is a quick glance to the most interesting results that you obtained with your work, followed by some suggestions.
Line 288: Their equipment is not in accordance with the regulations or they do not wear protective clothing made for forestry workers seeking the best combination of safety and optimal ergonomics in daily work. These aspects (as others in the following lines) were not treated before.
Line 311: Mechanical and physical risk prevention must be supported in small and medium-sized enterprises. ??? No prevention in small forestry enterprises?
Line 316: Even though the results of the analysis show a decrease in injury rate considering an increase in forest production, misinterpretation is a common problem when using statistical information, especially data related to self-employed. Not clear.
Please avoid citations in the conclusion chapter.
Author Response
Dear reviewer,
Thank you very much for the detailed, word-to-word proofreading. We tried to incorporate all suggestions into the paper. Due to specific conditions in Slovak forestry, some results cannot be explained in more detail. Therefore, external reviewers may find some facts unclear or even confusing.
Quotations in lines 116-117 are essential in terms of giving reasons for the selection of the appropriate research methodology.
Common leg injuries (bone fractures, open wound, etc.) are considered serious but not life-threatening. Vascular trauma or limb amputation occur only rarely in forestry jobs.
It is very difficult to find authors dealing with this issue in similar conditions and to compare their results to those gathered in our research. Most works cited in the part - discussion are aimed at the jobs and injuries occurring in the forest sector.
We would like to kindly ask you to point out what exactly you find unclear or confusing in Discussion and Conclusion section.
Following is the point by point reply:
Overall
The paper is not well structured and a deep revision is necessary. Some parts must be completely rewritten and some words are not appropriate. Also the structure of the sentences must be revised.
The number of the decimals must be uniform along all the text.
Was changed, in table 3, also in text
Table and figure inside titles must be avoided.
Titles have been removed
The references must be shortened (too much repetitions and/or not pertinent papers)
References were edited
Introduction
Incorrect words are present at lines 32, 34, 37
Was changed
You start referring to the risks in forestry sector (lines 33-42), then you cite not pertinent works ([7-10],line 45).
Was Changed, we use the properly references
Lines 46-49: not clear, please explain better
Was changed
Line 52: local vibration. What is this?
Was changed only on factor of „vibration“
In the introduction you mix the forest sector with other human activities, increasing the confusion in the mind of the reader. Beside this, the goal statement of the paper is not clear.
we have more explicitly focused on the forestry sector
Material and method
Lines 116-117: reduce the 7 citations to one, max 2: this is not a work on epidemiology.
Quotations in lines 116-117 are essential in terms of giving reasons for the selection of the appropriate research methodology.
Line 121: write better. What does it mean m3 (N)?
Was changed on unit 1,000 m3 (in all text)
If data gathered in the year 2017 cannot be included in the analysis, this year must be avoided since the beginning (line 71).
Was changed – data time series 2000-2016 (in all text)
Line 128: As the Poisson distribution assumes that, the mean is greater than zero, the function of prognostic variables is often limited and the values of a function are greater than zero. Not clear.
Line 130: Following logarithmic-linear model is usually used to express the rate λ as a function of a set of prognostic variables X1,. . . . . Xp. Please, explain better.
Was changed
Line 134: The acronym of the generalized linear models is GLMs, not GLIMs.
Was changed
Line 136: This term is called the “offset”. [27-28]. One citation is sufficient.
we left two citations
Line 138: invariable coefficient … What does it mean?
Was changed
Line 140: wood in thousand.m3. ????? Not scientific
Was changed on unit 1,000 m3 (in all text)
Line 140: However, the risk of part-time workers cannot be compared to the risk of full-time workers. And then?
Line 142: The variance and the mean are equal is a condition to select Poisson distribution for the number of injuries (d). ??? Not clear.
Was changed
Line 143. Multiple linear regression with homogeneity of variances is based on the distribution. Not clear
Line 144-149: Absolutely not clear. Explain better
This explains the statistics in the cited works
Results
Line 160: Timber harvesting is considered the most risky phase of the process of harvesting, processing and transport (Fig. 1). If this is a result of a data collection and not of a personal perception, the term ‘considered’ is not correct. Moreover, if I look the figure 1, I see the forest harvesting as the highest risky operation.
Was changed
Line 163: Timber harvesting phase is one of the riskiest phases. ??? Not clear. Moreover, it not correct.
Was changed
Line 164: 48 % of fatal injuries and 26% of serious injuries leading to permanent damage to health were recorded in the forest industry during the analysis period. ??? Again: the forest sector is or is not the objective of the investigation?
Was changed
Lines 170-178: to be moved in material and method.
We believe that this is more appropriate here since it is directly related to the results in Table 2
Line 189: More fatal injuries than was the expected value occurred during the phase of harvesting and skidding. Explain better
Was changed
Line 192: Due to the coefficient of association of rAB = 0.32 there is a moderate positive relationship. Very, very questionable…
Figure 2: Please move out the text of the title of the figure and explain it in the paper.
Was changed
Line 199: thousand.m3. Not scientifically correct.
Was changed on unit 1,000 m3 (in all text)
Line 203: In this model, it is supposed that the injury rate is the same for all forest workers regardless of the gender and age. Questionable sentence. It has been studied that in the forestry activities the injury rate is related to the two mentioned variables.
Figure 3: as written for figure 2.
Line 209 (Figure 3 caption): logarithmic transmission????
Was changed
Line 211: dependence variable: not used
Was changed
Table 3 is not well structured. Please avoid titles inside the tables and the figures, but move out in the text.
Line 213: Coefficients of creating the model????
Line 213-219: rewrite and explain better
simple regression results
Line 223: anthropometric data of the Slovak adult population has been increasing. What does it mean?
Was changed
Line 228: Injured body part in terms of operations?????
Was changed
Line 229: (48 % of all recorded injuries in forestry during the analysis period). Already written. Avoid repetitions.
Was deleted
Line 230: Following the results ????
Line 232: Leg injuries are not usually life-threatening. Don’t true. It depends.
Common leg injuries (bone fractures, open wound, etc.) are considered serious but not life-threatening. Vascular trauma or limb amputation occur only rarely in forestry jobs.
Line 229-238: Twisted sentences… difficult to understand…
Discussion
The discussion chapter is a comparison between the current work and others concerning the same topics. Here we find comments on other works (especially from line 246 to line 255), not linked to this paper. Moreover, some of the cited papers do not relate to the forest operations. There are two options: remove these citations or move them to the introduction chapter.
All the chapter is confused and must be completely rewritten.
It is very difficult to find authors dealing with this issue in similar conditions and to compare their results to those gathered in our research. Most works cited in the part - discussion are aimed at the jobs and injuries occurring in the forest sector.
We would like to kindly ask you to point out what exactly you find unclear or confusing.
Conclusion
Conclusions are too long and must be shortened, also for avoiding confusion.
Was changed
Line 277: Fatal injuries or serious injuries leading to permanent damage to health can occur, whereby, injuries are difficult to predict or avoid. Not understandable
Was changed
In the conclusion chapter you mention the danger when using chainsaws, but in the results you never cited the machines. Conclusion is a quick glance to the most interesting results that you obtained with your work, followed by some suggestions.
In Slovakia are timber-harvesting operations conducted mainly with chainsaws. Therefore, this statement at the conclusion.
Line 288: Their equipment is not in accordance with the regulations or they do not wear protective clothing made for forestry workers seeking the best combination of safety and optimal ergonomics in daily work. These aspects (as others in the following lines) were not treated before.
Line 311: Mechanical and physical risk prevention must be supported in small and medium-sized enterprises. ??? No prevention in small forestry enterprises?
Line 316: Even though the results of the analysis show a decrease in injury rate considering an increase in forest production, misinterpretation is a common problem when using statistical information, especially data related to self-employed. Not clear.
Due to specific conditions in Slovak forestry, some results cannot be explained in more detail. It is not scientific, but there are many more factors (objective and subjective). Therefore, external reviewers may find some facts unclear or even confusing. We would like to kindly ask you to point out what exactly you find unclear or confusing.
Please avoid citations in the conclusion chapter.
Was changed, but we left one reference.
Reviewer 2 Report
Dear Authors, The manuscript demonstrates important work in understanding the relationship between types of forestry work and injuries to various body parts. I believe that a broad audience exists among forestry safety and health professionals, clinical professionals, as well as academics that will be excited to read of this work. The manuscript reads well but needs minor revision to improve grammar in the introduction section. I have some recommended edits, questions, and comments offered below.
Comments and Editorial Feedback:
Page 1, line 22 – replace “mentioned” with “forestry harvesting”
Page 1, line 31-32 – The second sentence is awkward. I suggest the following: Long-term physical stress, strain, and associated pain and discomfort have a potential to damage tissue and progress to significant injury and disability. Chronic musculoskeletal conditions may also weaken individual capacity, degrade their performance, and increase risk for additional injury. (insert references)
Page 1, line 38 – The sentence is awkward. I suggest the following: In order to create healthy and safe working environments, processes, and conditions accident investigations can reveal information about causation that can be used to develop injury prevention and protection programs.
Page 2, line 44 – 45 end the first sentence after, …developing and developed. For example… and Sweden all report and record injuries and illness in small enterprises.
Page 2, line 46 – The sentence is awkward. I suggest the following: Many previous studies have focused on injury incidence and prevalence to underscore the need for great safety in the workplace. (insert references)
Page 2, line 57 – This is not correct. Authors have not investigated anthropometric factors related to injuries. This investigation evaluates injuries related to work phases or job tasks to anatomical body parts or regions. Please change this where needed. I suggest the following: The aim of the paper is to analyse work phases in forestry that may be associated with risk of injury by injured body part.
Author Response
Dear reviewer,
Thank you very much for the detailed, word-to-word proofreading. We tried to incorporate all suggestions into the paper.
Also thank you for your nice words about our work.
Page 1, line 22 – replace “mentioned” with “forestry harvesting”
Was changed
Page 1, line 31-32 – The second sentence is awkward. I suggest the following: Long-term physical stress, strain, and associated pain and discomfort have a potential to damage tissue and progress to significant injury and disability. Chronic musculoskeletal conditions may also weaken individual capacity, degrade their performance, and increase risk for additional injury. (insert references)
Sentence was changed
Page 1, line 38 – The sentence is awkward. I suggest the following: In order to create healthy and safe working environments, processes, and conditions accident investigations can reveal information about causation that can be used to develop injury prevention and protection programs.
Sentence was changed
Page 2, line 44 – 45 end the first sentence after, …developing and developed. For example… and Sweden all report and record injuries and illness in small enterprises.
Sentence was changed
Page 2, line 46 – The sentence is awkward. I suggest the following: Many previous studies have focused on injury incidence and prevalence to underscore the need for great safety in the workplace. (insert references)
Sentence was changed
Page 2, line 57 – This is not correct. Authors have not investigated anthropometric factors related to injuries. This investigation evaluates injuries related to work phases or job tasks to anatomical body parts or regions. Please change this where needed. I suggest the following: The aim of the paper is to analyse work phases in forestry that may be associated with risk of injury by injured body part.
Sentence was changed
Round 2
Reviewer 1 Report
You improved your work, but please, consider the comments. Thank you.
Equation 6: Please add the unit of measurement
Lines 122-123 are now useless, because you declared to work on 2000-2016 data.
Line 128: As the Poisson distribution assumes that, the mean is greater than zero, the function of prognostic variables is often limited and the values of a function are not in minus values. Please, work on the meaning of this sentence
You did not answer at this request of clarification
Line 130: Following logarithmic-linear model is usually used to express the rate λ as a function of a set
of prognostic variables X1,. . . . . Xp. Please, explain better.
Line 138: fixed value coefficient… a fixed value coefficient or fixed value coefficients?
Line 141: The variance and the mean are equal is a condition to select Poisson distribution for the number of injuries (d). You did not answer to my previous request. This sentence is not correct.
Line 144-149: you must improve your English in these sentences
You did not answer to this request of clarification:
Line 164: 48 % of fatal injuries and 26% of serious injuries leading to permanent damage to health
were recorded in the forest industry during the analysis period. ??? Again: the forest sector is or is not
the objective of the investigation?
Lines 170-178: to be moved in material and method. You can move the explanation of the Chi-square test of independence leaving here the results.
Line 192: Due to the coefficient of association of rAB = 0.32 there is a moderate positive relationship. Change in: The coefficient of association rAB = 0.32 indicates a low positive relationship.
Line 194: Injury rate in the forest industry over the years 2000 – 2017. No, 2016.
Line 196: Trend in the injury rate on 1,000 m3 in Slovak forestry sector ??? in the years 2000-2016.
Please correct Line 199: thousand.m3.
Line 208: after model (8)??? What does it mean?
Line 211: Coefficients the model (8) … please write correctly
Line 226: according to operations… perhaps occurred during forestry operations in …
Line 213-219: rewrite and explain better
simple regression results
I know, but it is not correctly explained
Line 227: The fact that the most common parts of the body affected by workplace accidents are legs (42.5 % of
all injuries) can be stated. Write better.
Line 236: 2000-2017. No. 2000-2016.
Discussion: if you agree that there are few works concerning this topic, you can write it in the text and remove useless comparisons as [35], for example, too much generic.
Line 257 2000 -2017… again…
In the conclusion chapter you mention the danger when using chainsaws, but in the results you never
cited the machines. Conclusion is a quick glance to the most interesting results that you obtained with
your work, followed by some suggestions.
In Slovakia are timber-harvesting operations conducted mainly with chainsaws. Therefore, this
statement at the conclusion.
If this is the situation, you must write it also in the introduction, otherwise the reader cannot understand why it appears only in the conclusion.
Author Response
Dear Reviewer, thank you again for the precise review of our work. We hope, that your detailed review can help improving our work.
You improved your work, but please, consider the comments. Thank you.
Equation 6: Please add the unit of measurement
Was changed
Lines 122-123 are now useless, because you declared to work on 2000-2016 data.
Was changed
Line 128: As the Poisson distribution assumes that, the mean is greater than zero, the function of prognostic variables is often limited and the values of a function are not in minus values. Please, work on the meaning of this sentence
You did not answer at this request of clarification
The sentence was changed: „As the Poisson distribution assumes that, the meaning value in distribution is greater than zero, this function of prognostic variables is often limited by the values of a function are greater than zero“
Line 130: Following logarithmic-linear model is usually used to express the rate λ as a function of a set
of prognostic variables X1,. . . . . Xp. Please, explain better.
The sentence was changed: „Following logarithmic-linear model is usually used to modeling the injury rate λ as a function of a set of prognostic variables X1, ......Xp.“
Line 138: fixed value coefficient… a fixed value coefficient or fixed value coefficients?
Is only one fixed value coefficient in this model.
Line 141: The variance and the mean are equal is a condition to select Poisson distribution for the number of injuries (d). You did not answer to my previous request. This sentence is not correct.
The sentence was changed: If the variance and the mean are equal, is the condition to select Poisson distribution for the number of injuries (d) fulfilled.
Line 144-149: you must improve your English in these sentences
You did not answer to this request of clarification:
Was changed
Line 164: 48 % of fatal injuries and 26% of serious injuries leading to permanent damage to health
were recorded in the forest industry during the analysis period. ??? Again: the forest sector is or is not
the objective of the investigation?
Yes the forest sector is primary objective of the investigation. Now it is also mentioned in introduction (line 60)
Lines 170-178: to be moved in material and method. You can move the explanation of the Chi-square test of independence leaving here the results.
Was moved in material and method – chapter 2.2
Line 192: Due to the coefficient of association of rAB = 0.32 there is a moderate positive relationship. Change in: The coefficient of association rAB = 0.32 indicates a low positive relationship.
Was changed
Line 194: Injury rate in the forest industry over the years 2000 – 2017. No, 2016.
Was changed
Line 196: Trend in the injury rate on 1,000 m3 in Slovak forestry sector ??? in the years 2000-2016.
Was changed
Please correct Line 199: thousand.m3.
Was corrected
Line 208: after model (8)??? What does it mean?
Was changed
Line 211: Coefficients the model (8) … please write correctly
Was corrected
Line 226: according to operations… perhaps occurred during forestry operations in …
You think the line 238 in figure capture? Was changed
Line 213-219: rewrite and explain better
simple regression results
I know, but it is not correctly explained
Was changed
Line 227: The fact that the most common parts of the body affected by workplace accidents are legs (42.5 % of
all injuries) can be stated. Write better.
Was changed
Line 236: 2000-2017. No. 2000-2016.
Was corrected
Discussion: if you agree that there are few works concerning this topic, you can write it in the text and remove useless comparisons as [35], for example, too much generic.
Comparison with multiple works may seem generic, but it is important to demonstrate that the subject has been developed by several authors under different conditions.
Line 257 2000 -2017… again…
Was changed
In the conclusion chapter you mention the danger when using chainsaws, but in the results you never
cited the machines. Conclusion is a quick glance to the most interesting results that you obtained with
your work, followed by some suggestions.
In Slovakia are timber-harvesting operations conducted mainly with chainsaws. Therefore, this
statement at the conclusion.
If this is the situation, you must write it also in the introduction, otherwise the reader cannot understand why it appears only in the conclusion.
We mentioned about this in introduction and also in the results chapter – Was changed